# UAV Time-Domain Electromagnetic System and a Workflow for Subsurface Targets Detection

Kang Xing [1,2,3], Shiyan Li [4], Zhijie Qu [1,2,3], Miaomiao Gao [1,2,3], Yuan Gao [1,2,3] and Xiaojuan Zhang [1,2,*]

[1] Aerospace Information Research Institute, Chinese Academy of Sciences, Beijing 100094, China; xingkang19@mails.ucas.ac.cn (K.X.); quzhijie20@mails.ucas.ac.cn (Z.Q.); gaomiaomiao20@mails.ucas.ac.cn (M.G.); gaoyuan21@mails.ucas.ac.cn (Y.G.)
[2] Key Laboratory of Electromagnetic Radiation and Sensing Technology, Chinese Academy of Sciences, Beijing 100190, China
[3] School of Electronic, Electrical and Communication Engineering, University of Chinese Academy of Sciences, Beijing 100049, China
[4] Tianjin Navigation Instruments Research Institute, Tianjin 300131, China; lishiyan18@mails.ucas.ac.cn
[*] Correspondence: xjzhang@aircas.ac.cn; Tel.: +86-10-5888-7276

**Abstract:** The time-domain electromagnetic (TDEM) method is acknowledged for its simplicity in setup and non-intrusive detection capabilities, particularly within shallow subsurface detection methodologies. However, extant TDEM systems encounter constraints when detecting intricate topographies and hazardous zones. The rapid evolution in unmanned aerial vehicle (UAV) technology has engendered the inception of UAV-based time-domain electromagnetic systems, thereby augmenting detection efficiency while mitigating potential risks associated with human casualties. This study introduces the UAV-TDEM system designed explicitly for discerning shallow subsurface targets. The system comprises a UAV platform, a host system, and sensors that capture the electromagnetic response of the area while concurrently recording real-time positional data. This study also proposes a processing technique rooted in robust local mean decomposition (RLMD) and approximate entropy (ApEn) methodology to address noise within the original data. Initially, the RLMD decomposes the original data to extract residuals alongside multiple product functions (PFs). Subsequently, the residual is combined with various PFs to yield several cumulative sums, wherein the approximate entropy of these cumulative sums is computed, and the resulting output signals are filtered using a predetermined threshold. Ultimately, the YOLOv8 (You Only Look Once version 8) network is employed to extract anomalous regions. The proposed denoising method can process data within one second, and the trained YOLOv8 network achieves an accuracy rate of 99.0% in the test set. Empirical validation through multiple flight tests substantiates the efficiency of UAV-TDEM in detecting targets situated up to 1 m below the surface. Both simulated and measured data corroborate the proposed workflow's effectiveness in mitigating noise and identifying targets.

**Keywords:** time-domain electromagnetic; subsurface target detection; unmanned aerial vehicle; denoising; YOLOv8

## 1. Introduction

In the past few decades, the electromagnetic induction (EMI) method has gained wide attention for its non-destructive and convenient advantages due to the urgent need for underground detection [1,2]. Scholars have developed various detection systems, including aerial [3–5], semi-airborne [6–8], and cart-mounted [9,10], and have also designed different detection devices to suit various scenarios, such as urban construction [11], archaeology [12,13], and pollution detection [14,15]. However, for detecting small areas like undulating terrains, areas with vegetation cover, and hazardous regions with unexploded ordnances, currently available detection devices still need to be improved regarding their

cost, detection efficiency, and construction safety [16,17]. The rapid advancement of unmanned aerial vehicle (UAV) technology has introduced the UAV-carried EMI method as a practical solution for the issues above.

UAV-carried EMI systems are similar to AEM (airborne electromagnetic method) systems and are classified into frequency domain electromagnetic methods (FEM) and time-domain electromagnetic methods (TDEM) based on the acquisition method [18]. The FEM is identified by capturing the secondary field signal while transmitting the primary field. The TDEM captures the secondary field after switching off the primary field. These methods are not fundamentally different and can be transformed into each other through a Fourier transform.

Karaoulis et al. used a DJI Matrice 600 with a GEM-2 system and a CMD MiniExplorer system for their UAV-FDEM experiments, respectively [19,20]. Their study went through three stages, first verifying the feasibility of the UAV-mounted frequency domain system by testing the noise level, then optimizing the system for test flights, and finally verifying its reliability by three field tests that successfully mapped shallow groundwater saturation and surface water salinity. Li et al. developed the AFEM-3 system, utilizing the UAV frequency domain electromagnetic method for detecting shallow underground targets [21]. The AFEM-3 host system is mounted on a small hexacopter UAV platform, and with sensors suspended by a long rope, it avoids interference from the UAV. The transmitter module adopts sinusoidal pulse width modulation (SPWM) technology, enabling the generation of multi-frequency transmit waveforms with arbitrary frequency combinations. The transmitting coil uses reverse flux compensation technology to enhance the shielding of the primary magnetic field on the induction signal. During field testing, AFEM-3 detected all pre-buried targets successfully. Wang et al. utilized a self-developed TEM31 time-domain electromagnetic system on a small UAV to conduct subsurface resistivity profile imaging [22]. The system operates at a fundamental frequency of 12.5 Hz and features a transmitting coil size of 2 m × 2 m with six turns, a transmitting current of 15 A , and an equivalent receiving area of 3000 square meters. Qi et al. developed a UAV time-domain electromagnetic system for detecting unexploded ordnances [23]. The system utilizes a six-axis multi-rotor UAV with transmitting and receiving sensors suspended 15 m beneath it via a rope. The transmitting loop comes in a 2 m × 2 m or 4 m × 4 m configuration and can handle a maximum transmitting current of 10 A. The receiving coil is either 0.5 m × 0.5 m or 1 m × 1 m, with an equivalent receiving area 200 $m^2$. During the field experiments, the system effectively detected three targets, thereby proving the usefulness of the UAV-carried time-domain electromagnetic system for detecting unexploded ordnance.

Currently, there are limited studies on UAV-TDEM systems, largely at the validation stage and needing more mature products. Further optimization of the detection device and system usability is necessary. On the other hand, TDEM signal exhibits the characteristics of a wide frequency band, non-linear and non-stationary. Additionally, the signal is weak in the middle and late stages, which can be easily submerged in noise, making it challenging to suppress noise. Two primary TDEM denoising techniques are available, signal decomposition-based and machine learning-based, each involving distinct approaches and challenges. Deep learning techniques are mainly advantageous since they can learn features from data automatically, sidestepping dependence on subjective human-selected data, and have gradually become the core of contemporary machine learning methods [24–26]. Incorporating deep learning into TDEM data processing presents challenges that include high computational costs, necessary parameter adjustments, and variations in performance depending on architecture and training data size. Classical methods based on signal decomposition offer greater convenience and less computational intensity than machine learning-based approaches. Popular techniques in signal processing comprise wavelet transform (WT) [27], empirical mode decomposition (EMD) [28], variational mode decomposition (VMD), and other similar methods [29,30]. However, these methods are still deficient in parameterization, computational speed, and robustness.

This study presents a novel TDEM system employing a UAV platform and delineates a comprehensive data processing workflow. The system executes measurement, acquisition, and data processing utilizing UAV control software, electromagnetic system control software, and data post-processing software. It comprises a UAV platform and a host system with transceiver sensors. The host system integrates various components, including a WIFI module, a real time kinematic (RTK) module, a control module, transmitter module, and an acquisition module. The host computer transmits predetermined parameters to the control module, which then modulates the transmitter module's output waveform by sending a timing signal, generating a primary field in the transmitter sensor. This primary field induces a secondary field signal in the target, leading to an induced voltage in the receiver sensor. Following this, the acquisition module records the induced voltage after amplification, filtering, and analogue-to-digital conversion, subsequently storing the data. Upon completing the measurements, the data undergo further processing through the implemented workflow. The proposed workflow involves the application of the robust local mean decomposition (RLMD) and approximate entropy (ApEn) method for noise removal, followed by automatic target detection using YOLOv8 (You Only Look Once version 8). Initially, the acquired signal undergoes decomposition by RLMD into residuals and multiple product functions (PFs). Subsequently, the PFs are sorted by frequency, and the cumulative sums of residuals and varying numbers of PFs are generated. The approximate entropy of these sums is computed and utilized as a criterion for signal filtering during the denoising process based on a predetermined threshold. Ultimately, the denoised signal is fed into YOLOv8 for target detection.

## 2. Principle of Time-Domain Electromagnetic Detection

The TDEM operates on the foundational principle of electromagnetic induction, depicted in Figure 1. A stable primary field initially envelops the transmitting coil when a continuous current flows through it. This primary field dissipates upon abrupt cessation of the transmission current, prompting the underground target to generate a time-varying eddy current. This current elicits a fluctuating secondary field detected by the receiving module. Targets exhibit distinct physical characteristics, inducing eddy currents with varied amplitudes and attenuation rates, consequently eliciting diverse secondary fields. Hence, this differentiation makes the discernment of a target's physical properties feasible.

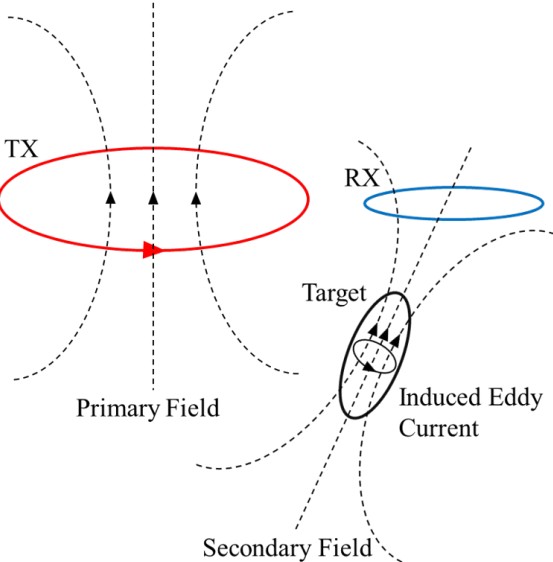

**Figure 1.** Schematic diagram of TDEM.

The formula for calculating the response of homogeneous formation in a half space has been given in [31]. For the central loop device, the side length of the rectangular transmitting coil is $L$, which has a stable current $I_0$, the earth conductivity is $\sigma$, and the

magnetic permeability in vacuum is $\mu_0$. At $t = 0$, the current steps off, and the eddy current induced in the formation diffuses downward with time. The vertical component of the secondary field $B_z$ and voltage of the receiving coil per unit area $V_z$ at the center of transmitting coil at $t$ is:

$$B_z = -\frac{\mu_0 I_0 \sqrt{\pi}}{2L}\left[\frac{3}{\theta L}e^{-\frac{\theta^2 L^2}{\pi}} + \left(1 - \frac{3\pi}{2\theta^2 L^2}\right)erf\left(\frac{\theta L}{\sqrt{\pi}}\right)\right] \tag{1}$$

$$V_z(t) = -\frac{dB_z}{dt} = \frac{I_0\pi^{\frac{3}{2}}}{\sigma L^3}\left[3erf\left(\frac{\theta L}{\sqrt{\pi}}\right) - \frac{2\theta L}{\pi}\left(3 + \frac{2\theta^2 L^2}{\pi}\right)e^{-\frac{\theta^2 L^2}{\pi}}\right] \tag{2}$$

where $\theta = \sqrt{\frac{\mu_0\sigma}{4t}}$, $erf(x) = \frac{2}{\sqrt{\pi}}\int_0^x e^{-\theta^2}d\theta$ represents the error function, $L$ represents the length of the transmit sensor.

It is assumed that there is a uniform conductor sphere with a radius of $a$, conductivity of $\sigma$, and permeability of $\mu$ below the transmitting coil. The transmission current which steps off at $t = 0$, secondary field generated by the eddy current in the sphere is:

$$V^s(t) = -\int_S \frac{\partial \boldsymbol{B}^s}{\partial t} \cdot d\boldsymbol{S} = -\int_S \frac{\mu_0}{4\pi r^3}\left[3\left(\frac{\partial \boldsymbol{m}}{\partial t} \cdot \boldsymbol{e}_r\right)\boldsymbol{e}_r - \frac{\partial \boldsymbol{m}}{\partial t}\right] \cdot d\boldsymbol{S} \tag{3}$$

where $r$ is the distance from sphere to the receiving coil, $\boldsymbol{e}_r$ is the unit position vector, and $\frac{\partial \boldsymbol{m}}{\partial t}$ is a partial derivative of the magnetic moment with time.

$$\frac{\partial \boldsymbol{m}}{\partial t} = \frac{2\pi}{\mu_0}\boldsymbol{B}^p\frac{\partial L}{\partial t} \tag{4}$$

$$\frac{\partial L}{\partial t} = -\frac{6a}{\mu_0\sigma}\sum_{s=1}^{\infty}\frac{q_s^2 e^{-\frac{q_s^2 t}{\mu_0\mu_r\sigma a^2}}}{q_s^2 + (\mu_r - 1)(\mu_r + 2)} \tag{5}$$

where $\boldsymbol{B}_p$ is the primary field generated by the transmitting coil in the sphere, and $q_s$ is the root of transcendental equation.

$$\tan q_s = \frac{(\mu_r - 1)q_s}{q_s^2 + (\mu_r - 1)} \tag{6}$$

Assume a transmitting coil with a side length of 1 m, a receiving coil with an effective area of 1 m$^2$, a transmission current of 1 A, an earth resistivity of 100 $\Omega\cdot$m. Another conductor sphere with a radius of 0.05 m, conductivity of $10^7$ S/m, and relative permeability of 200, is 1 m directly below the receiving coil, the secondary fields of the earth and the conductor sphere are received as shown in Figure 2.

From Figure 2, it can be seen that the amplitude of the secondary field response of the earth is larger than that of the conductive sphere in the early stage. In the middle and late stages, due to the large conductivity of the conductive sphere, the amplitude of the secondary field is much larger than that of the earth response. According to this, the underground abnormal body can be detected.

Under the same device conditions, the response of the conductor spheres with different conductivity and depths is shown in Figure 3.

As shown in Figure 4, the larger the conductivity, the smaller the early response of the sphere and the slower the decay of the secondary field.

When the burial depth is larger, the corresponding response is smaller. The trend of response decay is consistent for different burial depths, where the sphere responds with increasing distance in accordance with the attenuation law of the magnetic dipole field decreasing.

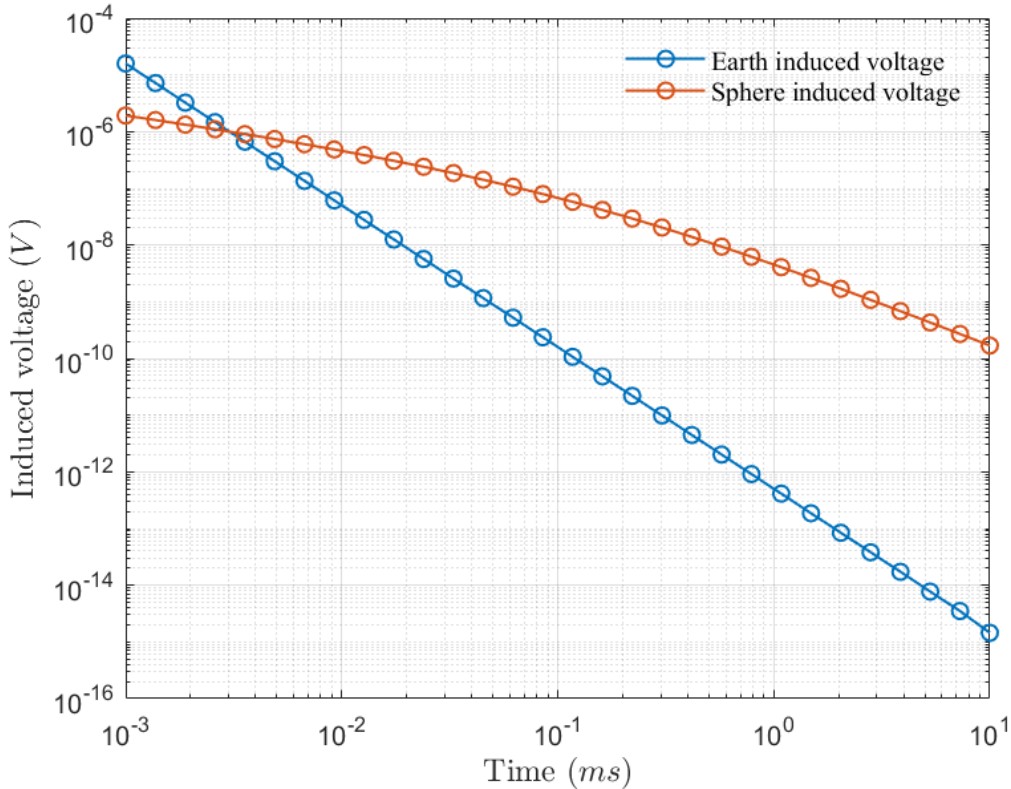

**Figure 2.** Comparison of the electromagnetic response of the earth and the conductor sphere.

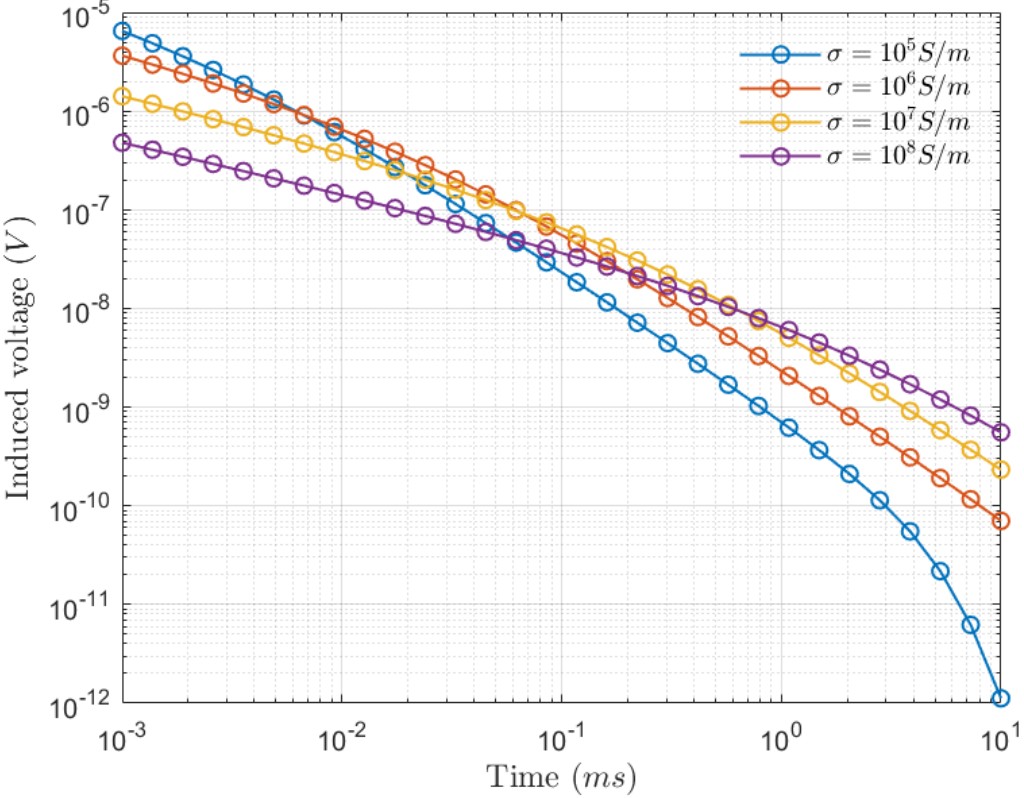

**Figure 3.** Comparison of the electromagnetic response of spheres with different conductivity conductors.

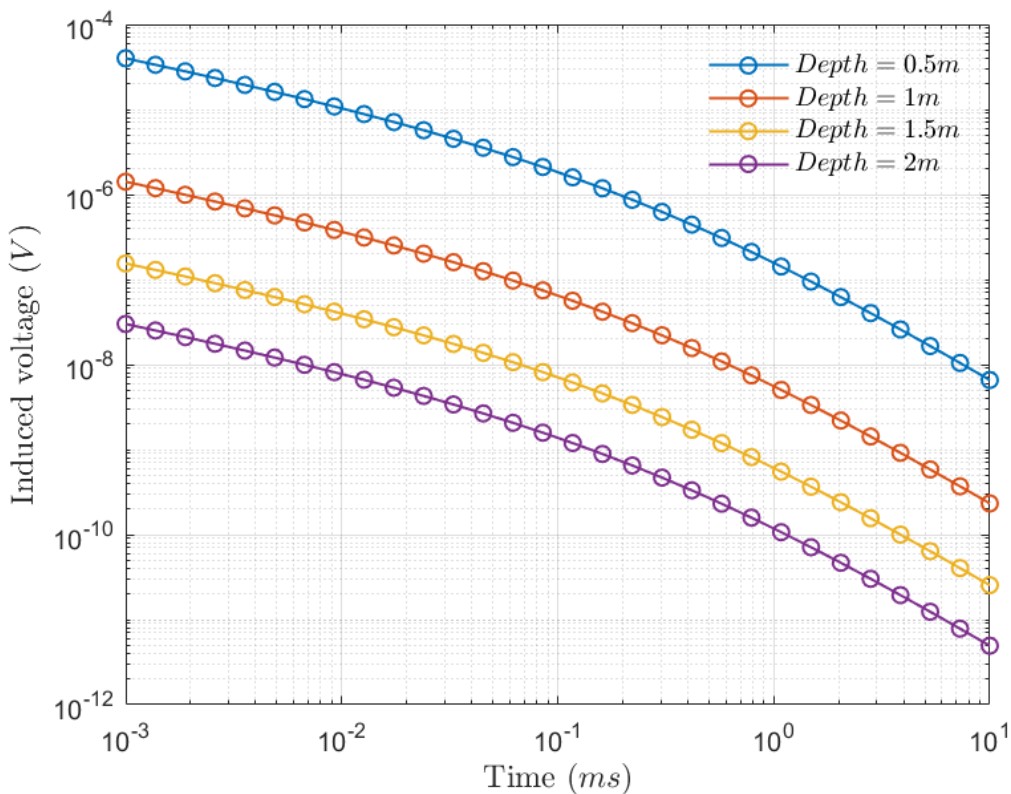

**Figure 4.** Comparison of the electromagnetic response of conductor spheres with different burial depths.

## 3. System

The UAV-TDEM system is mainly composed of a UAV platform, host system, and sensors, as shown in Figure 5.

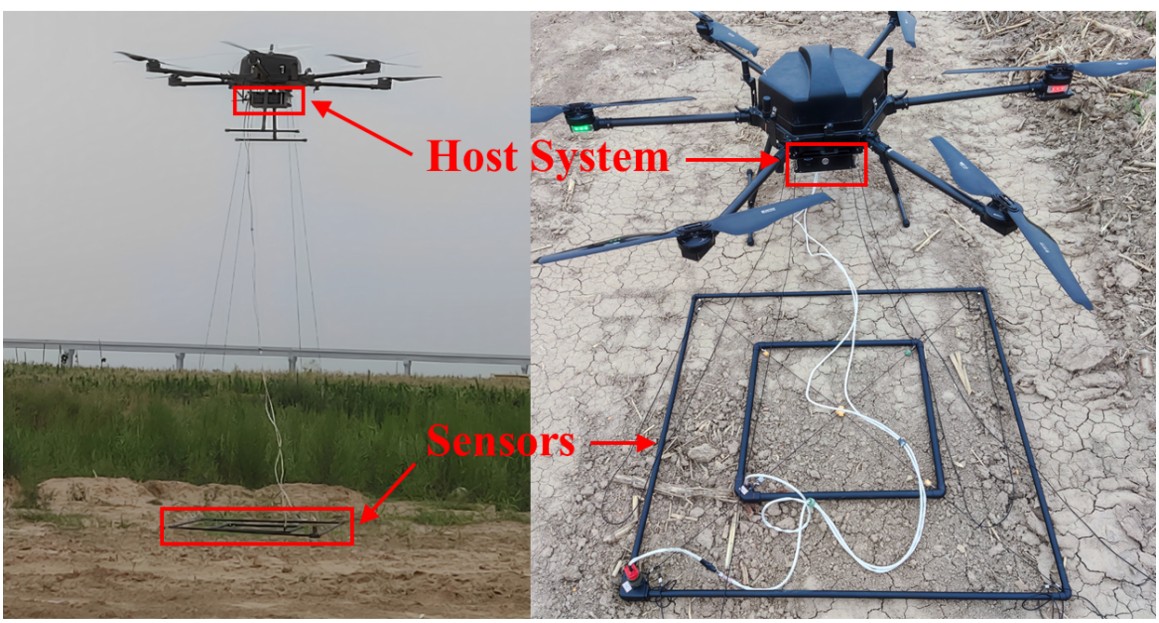

**Figure 5.** UAV-TDEM structure diagram.

### 3.1. UAV Platform

Our primary considerations encompass the drone's flight precision, duration, and terrain-following abilities. Choosing drones with minimal flight intervals and exceptional flight precision is essential for high detection accuracy. For detecting across large areas, drones that can carry heavier payloads and have extended flight durations are vital. Moreover, to ensure safe operation and seamless navigation in complex terrains, it is crucial to select drones equipped with sophisticated terrain tracking technology. After thorough comparative testing, the six-rotor drone from Sunward Technology Co., Ltd. (Zhuzhou, China) stood out as the preferred choice. This drone is notable for its maximum take-off load capacity of 13 kg and provides over 40 min of flight time when carrying a 7 kg load. It also supports a minimum route interval of 0.4 m, enhancing RTK functionality and enabling effective emulation of ground flight.

Four nylon ropes, each three meters long, are secured to the corners of the transmitting sensor, thereby suspending it directly underneath the drone. At this range, the noise caused by the drone's material or its motors is negligible compared to the response of shallow buried targets underground. The stability of the drone's flight significantly impacts the sensor's orientation, consequently influencing the quality of the gathered data. To limit noise interference caused by the drone's movements, we opted against using longer ropes. Additionally, the drone's flight speed and the interval between data collection strips directly affect the lateral resolution. In this case, the drone flies north–south at a speed of one meter per second, with an interval of 0.5 m.

### 3.2. Host System

The host system is composed of a control module, transmitting module, and acquisition module. The schematic diagram of the system architecture is shown Figure 6.

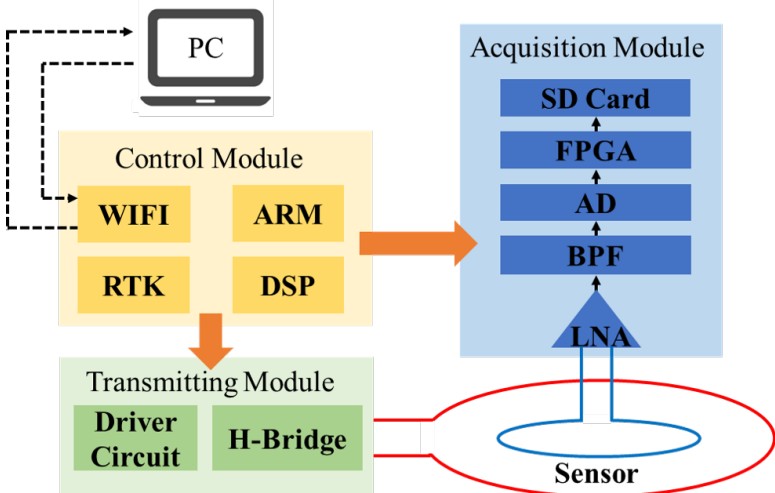

**Figure 6.** Schematic diagram of the mainframe system.

The control module facilitates communication with the host computer via WIFI, enabling parameter adjustments such as the transmit frequency setting. Simultaneously, it processes positional data for precise time synchronization and administers the timing signals for transmission and acquisition.

Within the transmitting module, the driving circuit operates by controlling the switch of four metal oxide semiconductor field effect transistor (MOSFET) gates based on the timing signal provided by the control module. This action establishes either a clockwise or anti-clockwise path in the transmitting coil, allowing the power supply to charge the transmitting coil. Upon completion of the charging process, the MOSFET gates are directed by the timing signal to deactivate the path, initiating the emission of a field. When the frequency of the emitted current is 25 Hz, during the 0–10 ms interval, the transmission

timing control circuit, detailed in Figure 7, activates switches Q1 and Q4. This action establishes a pathway through the red line in Figure 7a, resulting in the emission of a positive current. Conversely, in the 20–30 ms interval, switches Q2 and Q3 are activated to transmit a negative current in Figure 7b. Typically, the TDEM system employs the transmission of bipolar currents, as illustrated in Figure 8, which depicts the current waveform.

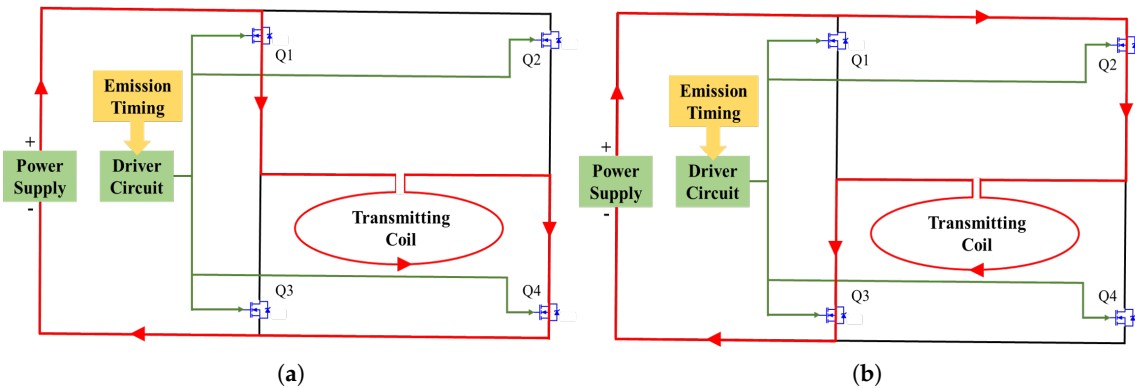

(**a**)                                                                        (**b**)

**Figure 7.** Schematic diagram of transmitting circuit. (**a**) Positive current; (**b**) negative current.

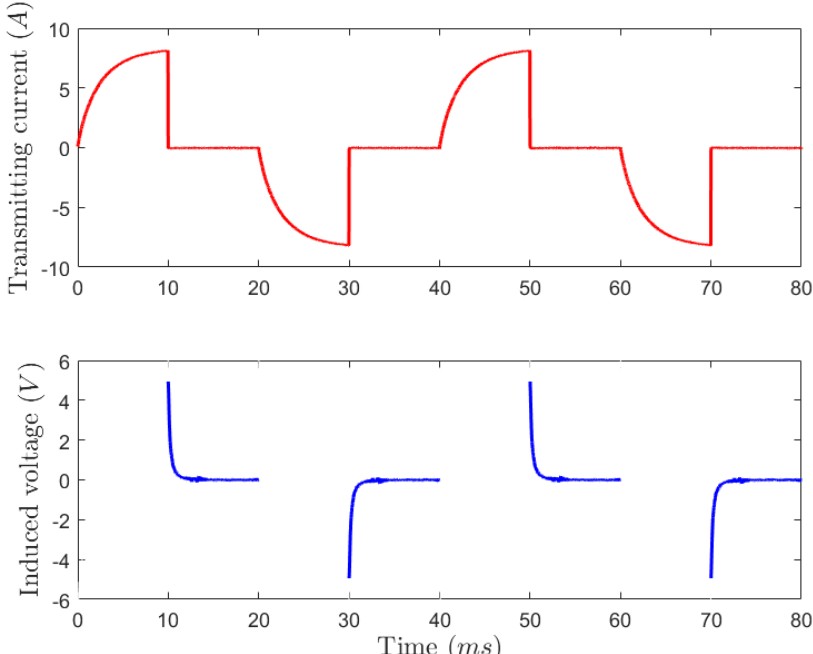

**Figure 8.** Measured transmission current and receive signal.

Upon the cessation of the transmission current, the abnormal body induces a voltage signal at both terminals of the receiving coil, eliciting a secondary field. The acquisition module is designed to measure the voltage difference across the receiving coil. This voltage difference is then subjected to a sequence of processing steps: amplification using a low-noise amplifier, filtration through a low-pass filter, and analog-to-digital conversion. The processed signal is subsequently recorded onto the host system's SD card via a field programmable gate array (FPGA) interface. Parallel to that, the data are transmitted to the host computer through a WIFI module, facilitating real-time display of the collected reception waveforms. Figure 8 illustrates the transmitted and collected signals.

*3.3. Sensors*

The sensor comprises a transmitting coil and a receiving coil. The transmitting coil features a side length of 1 m, comprising 15 turns, and operates at a peak transmitting current of 8 A. Conversely, the receiving coil measures 0.5 m in side length, comprises 20 turns, and possesses a bandwidth of 140 kHz.

Key parameters governing the transmitting coil are the transmitting magnetic moment and the turn-off time. Increasing the transmitting magnetic moment facilitates deeper detection capabilities, while reducing the turn-off time enhances the accuracy of acquired secondary field signals. Regarding the receiving coil, pivotal parameters include the effective area and bandwidth. A larger effective area directly correlates with a heightened amplitude of the primary captured secondary field signal. Additionally, a wider bandwidth diminishes the influence of the receiver coil's transfer function on the secondary field signal.

**4. Workflow**

The data collected by the system frequently include various types of noise, such as random sky noise, power frequency noise, and motion noise. With the nonlinear and non-stationary characteristics of TDEM signals, meaningful information is challenging. Considering the shortcomings of current signal decomposition-based methods in adaptivity, computational speed, and robustness, this paper introduces an innovative approach employing RLMD and ApEn. RLMD decomposes the original data to separate the residuals and multiple PFs. These are then recombined in various ways to create several cumulative sums. The approximate entropy of these sums is calculated, and signals are filtered based on a pre-set threshold. The specific process is shown in Algorithm 1. This technique eliminates noise from the signal, enhancing the signal-to-noise ratio and laying a solid foundation for precise target detection.

---

**Algorithm 1** TDEM signal denoising with RLMD-ApEn

---

**Require:** TDEM Signal $x(t)$
**Ensure:** TDEM signal after denoising $x_{denoise}(t)$
  1: **while** $u_i(t)$ not constant or monotonic **do**
  2:    $x(t) = u_i(t)$
  3:    **while** $\lim_{n \to \infty} a_{1n}(t) \neq 1$ **do**
  4:      compute the local mean $m_i$ and local envelope $a_i$
  5:      produce local mean function and amplitude function $m(t)$ and $a(t)$
  6:      calculate $h(t) = x(t) - m(t), s(t) = h(t)/a(t)$
  7:    **end while**
  8:    calculate $a_1(t) = \prod_{q=1}^{n} a_{1q}(t), f_{PF1}(t) = a_1(t)S_{1n}(t)$
  9:    calculate $u_1(t) = x(t) - f_{PF1}(t)$
10: **end while**
11: decomposition results $x(t) = \sum_1^k f_{PFk}(t) + u_k(t)$
12: **for** $i = 1; i < k, i = i + 1$ **do**
13:    **if** i==1 **then**
14:      $R_1 = u_k(t) + f_{PF1}(t)$
15:    **else**
16:      $R_i = R_{i-1} + f_{PFi}(t)$
17:    **end if**
18:    calculate the ApEn $A_i$ of $R_i$
19:    **if** $A_i \leq$ threshold **then**
20:      $x_{denoise}(t) = R_i$
21:    **else**
22:      $x_{denoise}(t) = R_{i-1}$
23:    **end if**
24: **end for**

---

*4.1. RLMD*

Local mean decomposition (LMD) is a time-frequency signal processing method proposed by Smith [32]. It can adaptively modulate the multi-component AM-FM signal into a series of PFs. Each PF is the product of the envelope signal and the FM signal, which can be regarded as a single-component AM-FM signal. For a given signal $x(t)$, we first find all the local extremum points of the signal, and calculate the local mean value m of two continuous extremum points $n_i$, $n_{i+1}$ and the envelope estimation value a of the local extremum points. The calculation formula is as follows :

$$\begin{cases} m_i = \dfrac{n_i + n_{i+1}}{2} \\ a_i = \dfrac{|n_i - n_{i+1}|}{2} \end{cases} \tag{7}$$

in which $n_i$ is the $i$th extreme point, $i = 1, 2, ..., K$ and $K$ donates the total number.

The local mean function $m(t)$ and the envelope function $a(t)$ are obtained by smoothing $m_i$ and $a_i$. The local mean $m(t)$ is removed from the original signal, and the zero mean signal $h(t)$ is obtained.

$$h(t) = x(t) - m(t) \tag{8}$$

The frequency modulation signal $s(t)$ is obtained by dividing $h(t)$ by $a(t)$.

$$s(t) = \frac{h(t)}{a(t)} \tag{9}$$

Repeat the above steps to obtain a pure FM signal $S_{1n}(t)$, such that $\lim_{n \to \infty} a_{1n}(t) = 1$. The envelope estimation in the iterative process is multiplied to obtain the envelope signal.

$$a_1(t) = \prod_{q=1}^{n} a_{1q}(t) \tag{10}$$

The envelope signal is multiplied by the frequency modulation signal $S_{1n}(t)$ to obtain the first PF.

$$f_{PF1}(t) = a_1(t)S_{1n}(t) \tag{11}$$

The new signal $u_1(t)$ is obtained by subtracting the first PF from the original signal, and it is used as the new original signal to repeat the above steps for k times until $u_k(t)$ is a constant or monotone function. Finally, the original signal is decomposed into several PFs and a residual component $u_k(t)$.

$$x(t) = \sum_{1}^{k} f_{PFk}(t) + u_k(t) \tag{12}$$

Although LMD can be adaptively decomposed according to signal characteristics, it has the problem of end effect and mode mixing. RLMD optimizes the boundary conditions, envelope estimation, and screening stop conditions, which effectively alleviate the above problems [33,34]. The improvement measures are as follows.

1.  The mirror expansion algorithm is introduced to deal with the boundary conditions.
2.  Automatically determine the fixed subset size of the envelope signal according to the following formula.

$$\mu_s = \sum_{i=1}^{N} L_i S_i \tag{13}$$

$$\delta_s = \sqrt{\sum_{i=1}^{N} (L_i - \mu_s)^2 S_i} \tag{14}$$

$$\lambda = odd(\mu_s + 3 \times \delta_s) \tag{15}$$

where $L_i$ donates the step length of the local mean; $S_i$ donates the product function; $\mu_s$ is the mean of $L_i$; $\delta_s$ is the standard deviation of $L_i$; $odd()$ represents the odd number whose output value is greater than or equal to the input value.

3. According to the evaluation function obtained by three consecutive iterations, it is judged whether to stop the screening, and the evaluation function is as follows:

$$RMS(z(t)) = \sqrt{\frac{1}{N} \sum_{t=1}^{N} (z(t))^2} \tag{16}$$

$$EK(z(t)) = \frac{\frac{1}{N} \sum_{t=1}^{N} (z(t) - \bar{z})^4}{\left( \frac{1}{N} \sum_{t=1}^{N} (z(t) - \bar{z})^2 \right)^2} - 3 \tag{17}$$

$$f = RMS(z(t)) + EK(z(t)) \tag{18}$$

where $z(t) = a(t) - 1$ represents zero baseline envelope signal, and $N$ donates the number of signal samples.

The Figure 9 displays the PF, AM, and FM signals derived from the RLMD decomposition of the simulation data. Notably, the high-frequency PFs exhibit higher noise elements, whereas the remaining low-frequency PFs comprise valuable information.

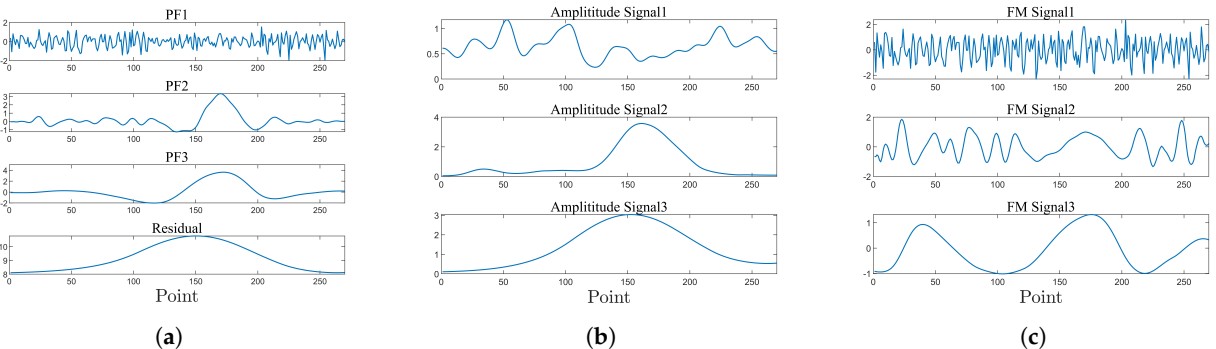

(**a**) (**b**) (**c**)

**Figure 9.** PFs, AM, and FM signals after RLMD decomposition. (**a**) PFs; (**b**) AM signals; (**c**) FM signals.

### 4.2. ApEn

Filtering out noise-dominated PFs and retaining signal-dominated PFs is imperative. Hence, to aid in this discrimination process, we introduce approximate entropy.

Approximate entropy [35], initially proposed by Steven M. Pincus in 1991, serves as a metric for assessing the complexity of signal sequences. It measures the likelihood of generating new patterns within signals. A higher ApEn value signifies greater independence among the data, reduced repetitive patterns, and heightened randomness. Consequently, higher ApEn values indicate that the PF is noise-dominated and should be disregarded. The algorithm for approximate entropy is outlined as follows:

For a given sequence, $u = \{u(1), u(2), \ldots, u(N)\}$ of length $N$, non-negative integer $m$ and positive real number $r$, we define:

$$\begin{aligned} x(i) &= \{u(i), u(i+1), \ldots, u(i+m-1)\} \\ x(j) &= \{u(j), u(j+1), \ldots, u(j+m-1)\} \end{aligned} \tag{19}$$

then calculate

$$d[x(i), x(j)] = max_{k=1,2,\ldots,m} (|u(i+k-1) - u(j+k-1)|) \tag{20}$$

$$C_i^m(r) = (\text{number of } j \leq N - m + 1 \text{ such that } d[x(i), x(j)] \leq r) / (N - m + 1) \quad (21)$$

Finally, we calculate $\phi^m(r)$, and obtain the mathematical expression of approximate entropy.

$$\phi^m(r) = \frac{1}{N - m + 1} \sum_{i=1}^{N-m+1} \log C_i^m(r) \quad (22)$$

$$\text{ApEn}(m, r, N)(u) = \phi^m(r) - \phi^{m+1}(r) \quad (23)$$

The determination of the approximate entropy threshold aligns with the specific signal characteristics. During RLMD decomposition, multiple PFs are generated, and the approximate entropy of the reconstructed signal is computed following the summation of various PF quantities. If the approximate entropy of the reconstructed signal falls below the threshold, it signifies signal-dominated PFs. Conversely, an approximate entropy exceeding the threshold indicates a mixture of the recovered signal with noise-dominated PFs, necessitating the removal of the PFs to eliminate noise effectively.

### 4.3. YOLOv8

Deep learning-based algorithms for target detection are broadly classified into two categories: two-stage and one-stage target detection algorithms [36]. Two-stage algorithms typically generate multiple candidate regions within an input image and subsequently classify these regions. Notably, the R-CNN series exemplifies this category. While these algorithms boast high accuracy, they demand substantial storage space and exhibit slower target detection speeds. In contrast, one-stage detection algorithms eliminate the need for candidate region generation. Instead, they directly convolve the entire image, significantly accelerating training speeds and enhancing real-time performance. Popular algorithms like SSD and YOLO adopt this approach [37].

Since its inception in 2015, the YOLO algorithm has progressively evolved into the most widely utilized single-stage target detection algorithm. Its notable versions include YOLOv5 and YOLOv7. YOLOv8, introduced by Ultralytics in January 2023, represents a significant update. Building upon its predecessor's strengths, YOLOv8 enhances the backbone network structure, detection header, and loss function to improve detection accuracy. The network structure is detailed in [38].

For automated target detection, we incorporate YOLOv8n, considering its balance between processing speed and accuracy. The dataset comprises 1000 contour maps illustrating induced voltage across the measurement area. It includes 2600 anomaly markers encompassing diverse parameters like target position, attitude, size, and responses under varying signal-to-noise ratios (SNR), shown in Figure 10. The training set constitutes 70%, the validation set 20%, and the remaining 10% form the test set.

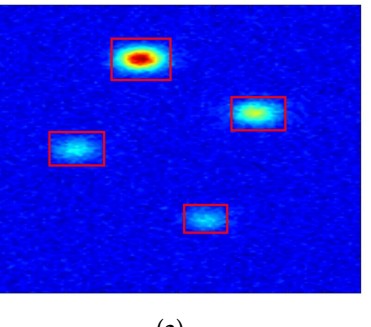
(a)

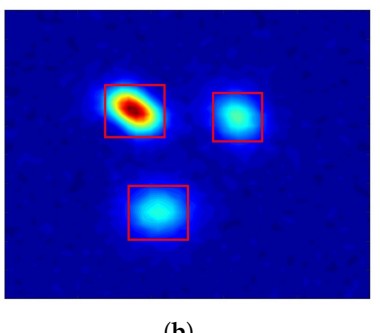
(b)

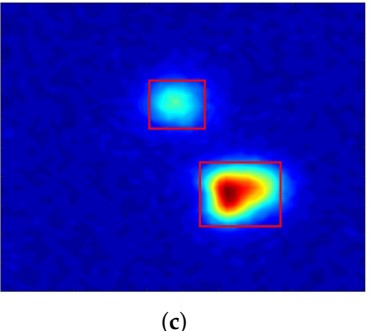
(c)

**Figure 10.** Partial samples and labelled anomalies in the dataset of YOLOv8. (**a**–**c**) show three maps generated by different targets with different azimuths and declinations, remanence, and locations.

The training parameters are defaulted except for batch is set to 16 and epoch is set to 200. The mean average precision (mAP) achieved by the trained network stands at 99.2%,

while on the test sets, it attains a 99.0% mAP. These results indicate the network's high accuracy in detecting anomalies.

## 5. Experimental Results

Experiments were conducted in Huayin City, Shaanxi Province, to assess the efficiency of the UAV-TDEM system. A test zone measuring 7 m × 10 m was designated, notable for its lack of vegetative cover and multiple undulations, each exhibiting a vertical variation of approximately 30 cm. The target parameters and locations are detailed in Table 1.

**Table 1.** Descriptions of targets used in the field experiment.

| Target ID | Material | Attitude | Size (cm) | Position (m) |
|:---:|:---:|:---:|:---:|:---:|
| 1 | Steel | Horizontal | [1] Φ15.5 × 2 | (1.8, 8.0, −0.4) |
| 2 | Steel | Vertical | Φ8 × 28 | (4.8, 4.6, −0.5) |
| 3 | Steel | Horizontal | Φ6 × 18.5 | (4.2, 3.0, −0.4) |
| 4 | Aluminum | Horizontal | 20 × 20 × 0.5 | (1.2, 6.5, −0.3) |
| 5 | Aluminum | Horizontal | 25 × 18 × 12 | (2.2, 6.1, −0.5) |
| 6 | Ferromagnetic | Vertical | Φ6 × 10 | (4.8, 9.3, −0.3) |
| 7 | Ferromagnetic | Horizontal | Φ10 × 20 | (5.8, 8.3, −0.4) |
| 8 | Ferromagnetic | Vertical | Φ15.5 × 25 | (5.3, 6.7, −0.5) |

[1] Φ means outer diameter.

The experimental and data processing procedure is shown in Figure 11. Initially, system installation and setup are completed, the detection area is selected, and flight paths are planned. The drone then autonomously surveys the area, collecting TDEM data stored on an SD card. In the second step, the data are transferred from the SD card to the computer for post-processing. The raw data undergo RLMD, generating a series of PFs. These PFs are then summed and the approximate entropy calculated. The resulting signals are filtered and interpolated into graphs. Finally, YOLOv8 is utilized for isoline detection and demarcation of anomalous areas.

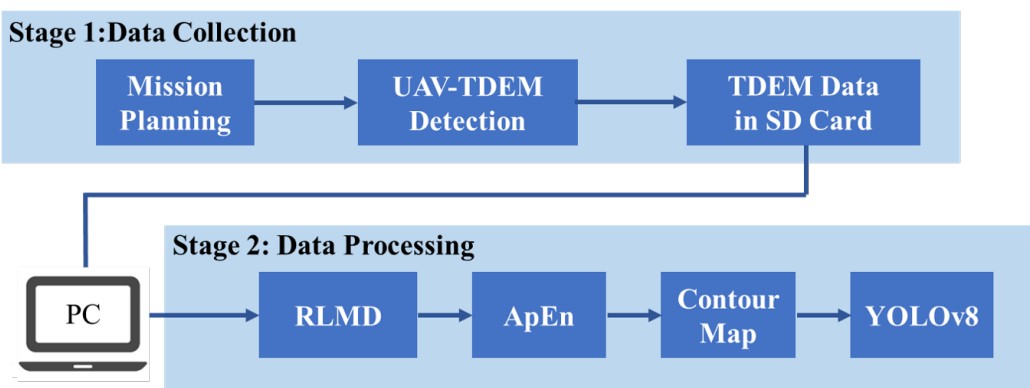

**Figure 11.** Schematic diagram of the experimental process and workflow.

Within the designated area, 14 flight routes were planned for the UAV, with a 0.5 m interval between adjacent survey lines. After trimming the turning paths at both ends, the flight routes are depicted in Figure 12. The red dot indicates the first point where the UAV enters the area. The irregularity in the mid-section of the routes is attributable to the drone's maneuvering, resulting in sensor oscillation. A flight speed of 1 m/s was set to reduce sensor movement.

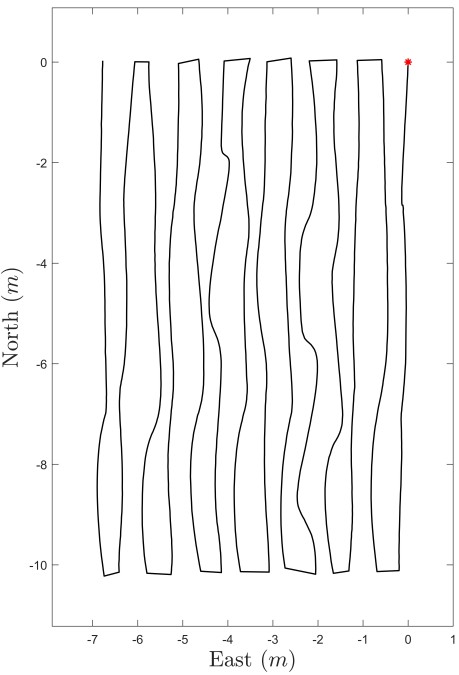

**Figure 12.** UAV flight routes after trimming the turning paths.

The raw data are presented in Figure 13, indicating that the target response of the eighth object is more significant due to its greater size and shallow depth. In contrast, the third and fifth responses are smaller and more susceptible to noise contamination. Consequently, determining the presence of multiple targets within the measurement area without prior knowledge is challenging. Figure 13b presents the result detected by YOLOv8. The original data's noise interference influences the detection outcomes, leading to misdetections and omissions.

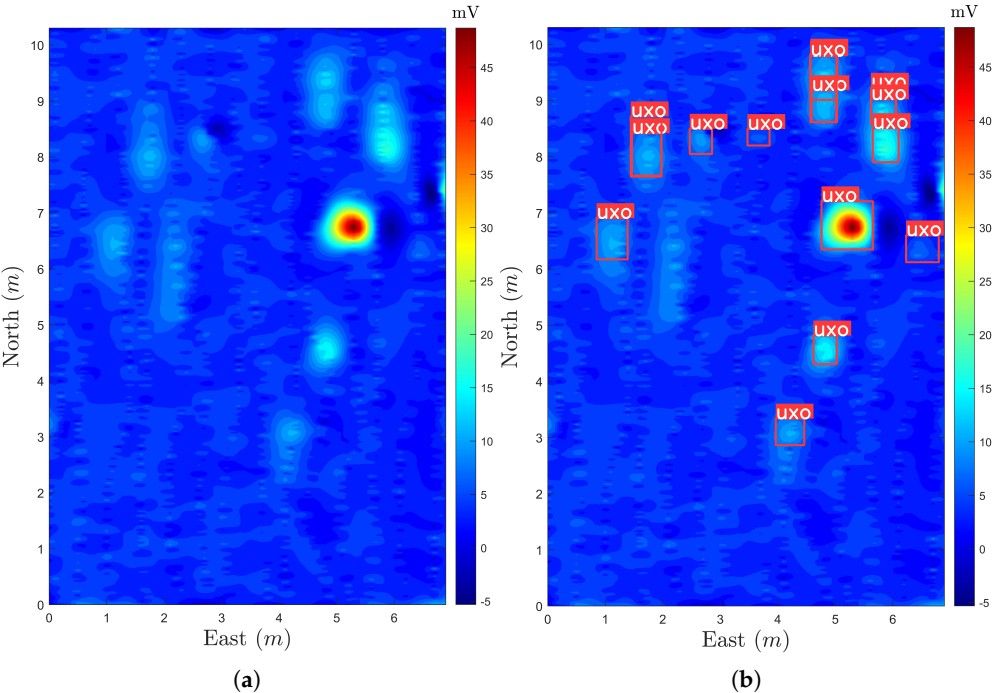

**Figure 13.** Electromagnetic response map of raw data and YOLOv8 detection results. (**a**) Electromagnetic response map of raw data; (**b**) YOLOv8 detection results for raw data.

The raw data are processed through the RLMD-ApEn, starting with adaptive decomposition using RLMD across 14 measurement lines, resulting in 14 sets of PFs and residuals. Figure 14 showcases the PFs and residuals obtained from the decomposition of measurement lines 2, 4, 6, and 10. Notably, the noise component predominantly dominates the high-frequency components in signals with low signal-to-noise ratios. Conversely, the signal component can overwhelm the high-frequency components in signals with higher signal-to-noise ratios.

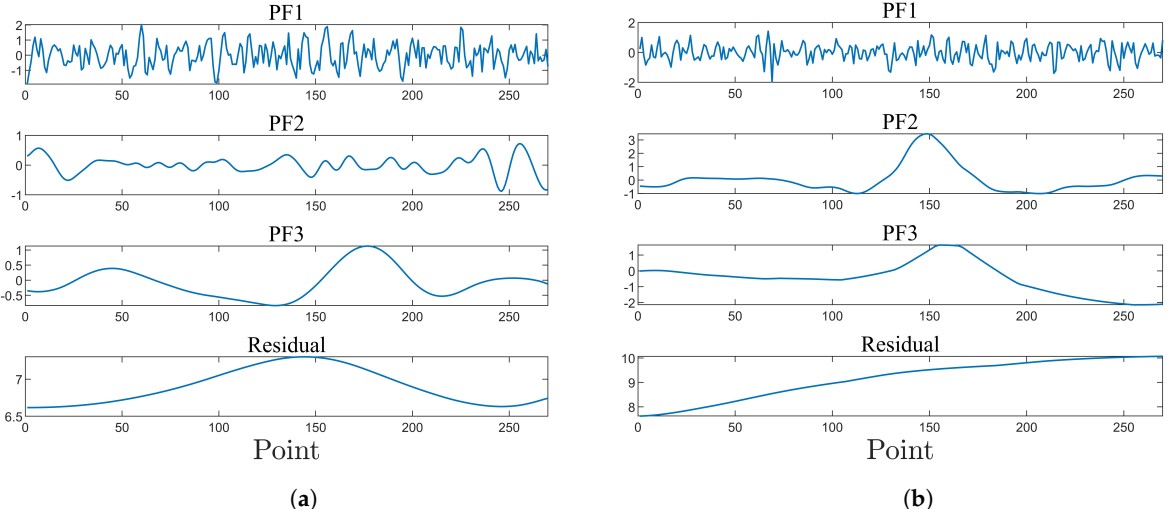

(a)

(b)

**Figure 14.** PFs obtained by RLMD decomposition of measurement lines 2 and 10. (**a**) PFs of line 2; (**b**) PFs of line 10.

Following this, the PFs are arranged from low to high frequencies, and the residuals are incrementally added to $PF_1$, $PF_1PF_2$, ..., and $PF_1PF_2PF_n$, generating n cumulative sums for separate approximate entropy calculations. Subsequently, the output signal is filtered based on the approximate entropy threshold, set at 0.3, derived from signal characteristics. Figure 15 illustrates the black and red lines denoting the approximate entropy of signals before and after correction for each measurement line. Notably, the approximate entropy of the first, second, seventh, eighth, ninth, thirteenth, and fourteenth measurement lines exceeded 0.3 before correction due to the inclusion of noise-dominated PFs during signal summation. However, post-correction, the approximate entropy of all measurement lines decreased to below 0.3, which means that the noise has been removed.

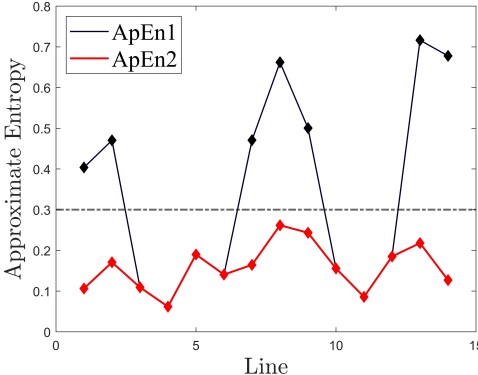

**Figure 15.** Approximate entropy before and after correction.

Figure 16 compares the proposed method and traditional approaches, ICEEMDAN, EWT, and VMD. The proposed method excels in noise removal, ensuring accurate signal restoration while preserving essential components for both low and high SNR signals.

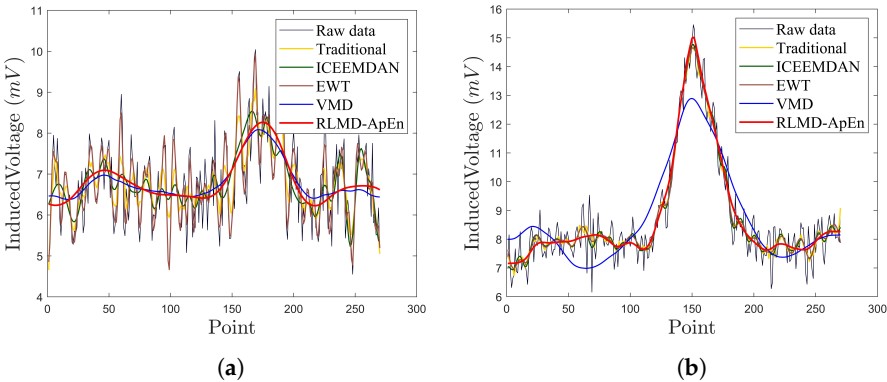

(**a**)                    (**b**)

**Figure 16.** Comparison of results of different methods. (**a**) line 2; (**b**) line 10.

Table 2 presents the time expended by various methods in processing the data from the survey area. Notably, both CEEMDAN and ICEEMDAN required a significantly longer processing time. The time cost of VMD was 4.7 times that of the proposed method. While Empirical Wavelet Transform (EWT) recorded the shortest processing time, it is comparable to the proposed method in terms of the order of magnitude of time expenditure.

**Table 2.** Time spent processing data by different methods.

| Methods | EWT | VMD | CEEMDAN | ICEEMDAN | RLMD-ApEn |
|---|---|---|---|---|---|
| Time costs (s) | 0.36 | 3.13 | 17.93 | 16.22 | 0.66 |

Figure 17 depicts the application of the proposed method to the original data. Figure 17a notably demonstrates a significant enhancement in image clarity, facilitating clear identification of anomalies. Conversely, Figure 17b showcases the outcomes of automated target detection using YOLOv8, successfully detecting all eight targets within the measurement area. These results affirm the effectiveness of the proposed method in noise reduction and the automated identification of target abnormalities.

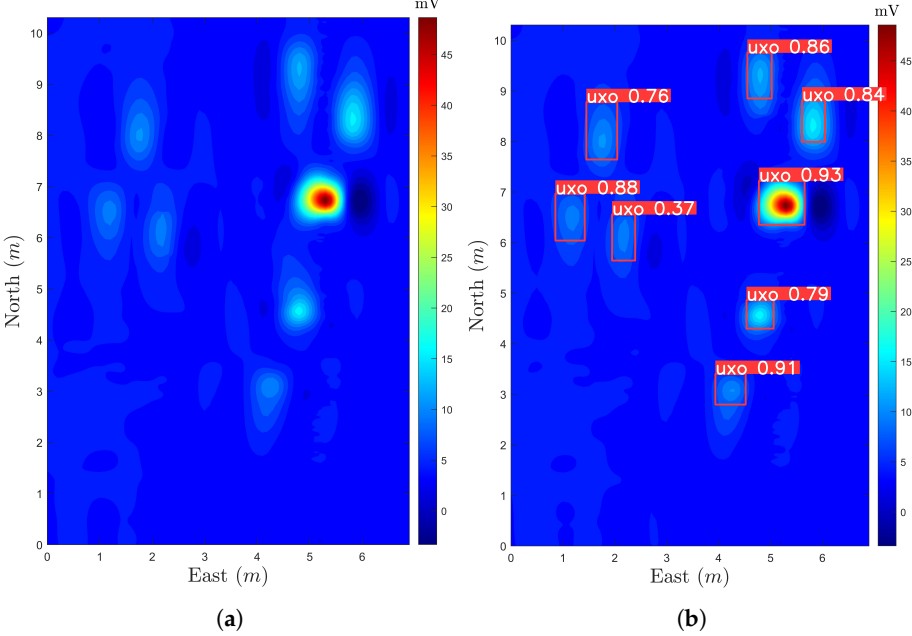

(**a**)                    (**b**)

**Figure 17.** Electromagnetic response map of processed data and YOLOv8 detection results. (**a**) Electromagnetic response map after RLMD-ApEn; (**b**) YOLOv8 detection results.

## 6. Discussion

This paper introduces a novel integration of TDEM systems with UAVs and delineates a comprehensive workflow. The system features a UAV platform and a host control system, including control, transmission, and reception modules, installed within the UAV's structure. The crucial transceiver sensor, tethered to the UAV via a nylon rope, facilitates a novel approach to electromagnetic surveying. Through extensive flight testing, the UAV-borne EM system has proven its efficacy in detecting near-surface targets. The workflow emphasizes two key processes: the initial denoising of raw data using the RLMD-ApEn method and subsequent target detection employing the advanced YOLOv8 algorithm. This dual-phase process, tested with both simulated and empirical data, has demonstrated remarkable efficiency and effectiveness in data processing and target identification.

### 6.1. Main Advantages

- The UAV-TDEM system is explicitly designed for shallow subsurface target detection, aimed at rapidly detecting and precisely locating underground targets. Compared to existing UAV-based time-domain electromagnetic systems, this system is more straightforward to operate, supports continuous mobile data acquisition, and is equipped with RTK module, enabling accurate recording of the locations of collection points.
- Unlike conventional handheld systems, TDEM-1, or vehicle-mounted systems TDEM-2, the system introduced in this paper represents an unmanned airborne time-domain electromagnetic method ideal for intricate terrains and hazardous zones. Operators only need to plan the flight route and set the collection parameters in the ground station software. The UAV-TDEM system can then autonomously conduct detection, significantly enhancing the system's detection efficiency and reducing the workload of operators.
- RLMD's adaptive signal decomposition and ApEn's proficient noise filtering ensure high-quality, clean data. This method is computationally efficient and robust, handling diverse signal types seamlessly.
- The integration of YOLOv8 allows for detecting varied target responses and achieves an impressive 99.2% accuracy rate, showcasing the reliability and effectiveness of automatic target detection.

### 6.2. Limits

- Compared to vehicle-mounted systems, UAV-based systems face limitations due to payload constraints. UAV-TDEM cannot generate larger transmission currents or utilize bigger sensors. These issues reduce the transmitted magnetic moment to some extent, consequently leading to a decrease in detection depth.
- The sensor's attachment via a nylon rope introduces challenges, such as oscillation during UAV movement and vulnerability to strong winds, affecting data consistency.
- Additionally, limited resolution hampers distinguishing between large individual targets, clusters of smaller ones nearby, or stacked formations.

Future research will concentrate on reducing errors caused by sensor altitude changes, utilizing improved hardware and sophisticated algorithms. Subsequent data collection will be directed towards refining methods for segmenting close-range target responses. Moreover, we plan to investigate inversion algorithms for the precise characterization and classification of targets, based on their magnetic moments and positions.

## 7. Conclusions

This paper delves into TDEM principles and techniques for near-surface target detection. It introduces a pioneering UAV-TDEM system and outlines a workflow for noise elimination and automated anomaly identification. A comprehensive overview of the UAV platform, a host system, and sensors used in the UAV-TDEM system is provided. The host system's control module manages the on–off state of the transmit circuit via timing generation, controlling the receive coil's response signal through AD control. The proposed noise

elimination method, RLMD-ApEn, involves adaptive decomposition of the input signal into residuals and PFs using RLMD. The output signal is constructed based on cumulative sums of residuals and varying PF numbers, adhering to the approximate entropy threshold. The denoising method is capable of processing data in under one second. Furthermore, a dataset of target responses was generated, and the YOLOv8 network was trained to facilitate the automatic recognition of these responses, achieving an accuracy rate of 99.0% in the test set. Field experiments corroborate the UAV-TDEM system's capability to detect all eight targets buried within 1 m of the ground. Implementing the proposed workflow renders target contours visible, affirming the system's detection prowess and validating the method's efficiency.

**Author Contributions:** Conceptualization, K.X. and X.Z.; methodology, K.X.; validation, K.X., S.L. and Z.Q.; formal analysis, X.Z.; investigation, K.X. and Y.G.; resources, K.X., S.L. and M.G.; data curation, K.X.; writing—original draft preparation, K.X.; writing—review and editing, X.Z.; visualization, Z.Q., M.G. and Y.G.; supervision, X.Z.; project administration, X.Z.; funding acquisition, X.Z. All authors have read and agreed to the published version of the manuscript.

**Funding:** This research was funded by the National Natural Science Foundation of China, grant number 61172017.

**Data Availability Statement:** Data are contained within the article.

**Conflicts of Interest:** The authors declare no conflict of interest.

## Abbreviations

The following abbreviations are used in this manuscript:

| | |
|---|---|
| UAV | Unmanned aerial vehicle |
| AEM | Airborne electromagnetic method |
| FEM | Frequency domain electromagnetic |
| TDEM | Time-domain electromagnetic |
| MOSFET | Metal oxide semiconductor field effect transistor |
| LMD | Local mean decomposition |
| RLMD | Robust local mean decomposition |
| ApEn | Approximate entropy |
| YOLO | You only look once |
| PF | Product function |
| FPGA | Field programmable gate array |
| RTK | Real time kinematic |
| SNR | Signal-to-noise ratios |
| mAP | Mean average precision |
| WT | Wavelet transform |
| EMD | Empirical mode decomposition |
| CEEMDAN | Complete ensemble empirical mode decomposition with adaptive noise |
| ICEEMDAN | Improved complete ensemble empirical mode decomposition with adaptive noise |
| VMD | Variational mode decomposition |

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
