# Peer review of "UAV Time-Domain Electromagnetic System and a Workflow for Subsurface Targets Detection"

_remotesensing, doi:10.3390/rs16020330_

Round 1
Reviewer 1 Report
Comments and Suggestions for Authors
Please have a look of attached file.

Reviewer 2 Report
Comments and Suggestions for Authors
The Time-Domain Electromagnetic (TDEM) method is a geophysical technique used to investigate the subsurface properties of the Earth. When applied to Unmanned Aerial Vehicles (UAVs), it allows for a more flexible and efficient data acquisition process compared to traditional ground-based surveys.
Lines 103-107: Please consider removing or rewriting as to emphasize the findings of the study, without mentioning the "chapters" of the article (not necessary, and not relevant).
Line 155: The sensor is connected to the drone via a rope - which type of rope? which is the length of the rope?
Also, some additional questions you should consider answering within the article text, sub-chapter 3.1: Depending on the lengths of the rope, the drone itself does not affect the data quality (source of noise)? Which is the speed of the drone during data acquisition? Are there preferred headings, like N-S vs W-E?
Lines 158-159: The 6-rotor drone manufactured by Sunward Technology Co., Ltd. emerged as the preferred choice after comparative testing. - comparative testing with which other UAV systems? Why was this one better?
5. Experimental Results: A short description of the test area would be helpful - was it a flat area? or undulating terrain / with vegetation cover? Please consider introducing a photo taken during test flight (drone vs terrain).
How does your method/workflow apply on undulating/rough terrain?
Line 369: sensor attitude changes
Round 2
Reviewer 1 Report
Comments and Suggestions for Authors
I appreciate the author's hard work to improve this manuscript. Significant changes from several reviewer comments succeeded in answering several things that needed to be added to the manuscript. However, several things from the author's comments have not been written in the manuscript. In comment 8, the author responded by adding a route statement from UAV sensing, but it is not yet visible in the manuscript. It is best to provide quantitative results in the abstract and conclusion section. And please pay attention to the spacing of some words, especially near the punctuation.
Best wishes.
